# Effects of Relative Positions of Defect to Inclusion on Nanocomposite Strength

**DOI:** 10.3390/ma15144906

**Published:** 2022-07-14

**Authors:** Jiaqin Wang, Vincent B. C. Tan

**Affiliations:** Department of Mechanical Engineering, National University of Singapore, Singapore 117575, Singapore; mpetanbc@nus.edu.sg

**Keywords:** nanocomposite, crack, mechanical strength, finite element simulation, epoxy

## Abstract

It is generally accepted that material inhomogeneity causes stress concentrations at the interface and thus reduces the overall strength of a composite. To overcome this reduction in strength, some groups experimented on coating the nanoinclusions with a layer of rubbery material, aiming for higher energy absorption. However, representative volume element (RVE) nanocomposite models, established with randomly distributed core–shell nanoparticles and single nanoparticle cells, show that the enhancement in strength observed in some experiments remains elusive computationally. By including a pre-existing crack in the matrix of the RVE, the stress concentration at the crack tip is reduced for cases where the nanoparticle and precrack are aligned away from the loading direction. This suggests that stress concentrations around inherent defects in materials can sometimes be reduced by adding nanoparticles to improve material strength. The effect is reversed if the crack and nanoparticle are aligned towards the loading direction. Parametric studies were also carried out in terms of the relative stiffness of the nanoparticle to the matrix and crack length. Validation tests were performed on 3D RVEs with an elliptical crack as the initial defect, and the results match with the 2D findings.

## 1. Introduction

Polymer nanocomposites have been an emerging field over the past three decades, ever since the first successful synthesis and characterization of a polymer/clay nanocomposite (PCN) with a nanoclay hybrid inside a nylon 6 matrix by the Toyota research group [1,2]. Shortly afterwards, clay-reinforced nanocomposites with epoxy as the matrix material were also successfully synthesized and characterized by Lan and Messersmith [3,4]. Polymer nanocomposites where nanofillers are dispersed inside a polymer matrix exhibit improved mechanical and thermal properties as compared to those of a neat matrix due to the addition of the nanofillers [5,6,7]. Compared with conventional fiber-reinforced composites where a relatively large filler amount is required, nanocomposites are able to achieve superior properties at a much lower content of nanofillers (usually a < 5% weight fraction) [5,6,7].

In most studies involving nanocomposites, tensile strength is compromised, while it enhances other mechanical properties such as Young’s modulus, regardless of the material systems involved [8,9,10,11,12,13]. To overcome this drawback, efforts were focused on simultaneously improving the stiffness and the strength of the nanocomposites [14,15,16,17,18,19]. When heterogeneity is introduced into a matrix, be it a harder or softer inclusion, it acts as a stress raiser and amplifies the stress experienced by the region near the inclusion and/or the interphase [20]. This was observed experimentally [8,11,21,22,23,24], proven theoretically [25,26], and simulated computationally [9,27]. In the extreme, if an inclusion has zero mass or stiffness, i.e., a void, an amplification effect of the maximal stress is found and experienced at the void boundary [25,26,28,29]. However, there exist some exceptions. He et al. [16] manufactured several nanocomposite specimens and found that, compared with pure matrix, nanocomposites with embedded nanosilica display higher values of tensile strength. Thitsartarn et al. [15], and Xia et al. [19] also incorporated rigid nanoparticles into the epoxy matrix to form a nanocomposite, and reported that the strength of the nanocomposite showed improvement over that of the pure epoxy specimen. Nevertheless, if the incorporated inclusion is of a more compliant and ductile material, such as rubber, there is feasibility to enhance the overall toughness and thus increase the energy absorption of the composite material [30,31].

Since most of the results show that the addition of the nanoscale inclusions is detrimental to the overall tensile strength of a nanocomposite, researchers have thought of the idea of coating the hard inclusion with a layer of rubber, and incorporated this core–shell particle into the composite. Some experimental work was performed on this topic [14,15,16,17,32,33,34]. A specific example is Liu et al. [14], who coated nanosilica particles with a copolymer layer of polylactic acid (PLA) and polycaprolactone (PCL) manufactured by the same group [35]. They were able to achieve concurrent improvements in the overall stiffness, tensile strength, and elongation to failure. The strength improvement was particularly promising in that, with only 0.5% and 1% of nanoparticles added, the nanocomposites were able to achieve 6% and 16% increases in strength, respectively. Nevertheless, not all work showed such desired outcomes. Quan et al. [36], and Wang et al. [13] fabricated polymer nanocomposites with core–shell nanoparticles as the inclusions and found that the addition of the nanoparticles reduced the tensile strength of the composite. Sun et al. [32,33] fabricated various types of core–shell nanocomposites. The tensile strengths improved for some cases and reduced for the rest. Moreover, Mao et al. [37] fabricated core–shell nanocomposites with different inclusion contents. Their results showed that, though the tensile strength increased when core–shell nanoparticles were first added (at 1%), it began to drop as the content increased, and eventually dropped below the strength of a pure matrix specimen where no nanoparticle is present.

This paper provides insight into why the results on nanocomposite strength are different among various research groups. For single-phase inclusion, though most reports agree that it acts as a stress raiser and thus lowers the composite strength, exceptions in experiments occur. Regarding a nanocomposite with core–shell nanoparticles, some results are very promising and others are not. This work proposes the idea of including an initially or naturally existing defect into the nanocomposite. Since initial defects are almost impossible to control in experiments, this work is carried out computationally. The initial defect was placed at various locations of a model, such that the effects of including this defect could be studied systematically.

## 2. Nanocomposite RVE Model with Core–Shell Nanoparticles

In order to have a clearer understanding of the reasons why there is no mutual consensus on the affected strength of nanocomposites, a numerical finite element (FE) model was built on the basis of the above-mentioned experimental work. For FE analyses, a representative volume element (RVE) is normally created to study the macroscopic properties of nanocomposites, since an RVE is able to explicitly represent the microscopic heterogeneous features such as the size, shape, and orientation of fillers. Mechanical properties such as Young’s modulus, tensile strength, and fracture toughness are normally the properties determined through RVE modeling. The computational setup is mainly based on the experimental works in [14,15] performed by the same research group. The diameter of the nanoparticle core was 12 nm, and the shell thickness was 3.4 nm. The nanoparticles were randomly distributed within the RVE, and the weight percentage of the nanoparticles was kept at 2%. The randomness of the nanoparticle distribution and size was created using an inhouse C++ program that utilizes a random number generator function. The randomly generated numbers were taken as the coordinates of the center of the nanoparticles. The coordinates were generated for one particle at a time. Each new set of coordinates had to be separated from all existing particle coordinates by a specified minimal distance. If not, the new coordinates were discarded, and another set of coordinates were generated. The particle size was referenced from the above experimental paper and is within the range of various experimental works. Other input parameters such as material properties are specified in Table 1. The model was established, and all simulations were performed through commercial finite element software Abaqus. The simulations were carried out on a PC with Intel Xeon(R) CPU E5-2697 v4 @ 2.30GHz, 128 GB RAM. The simulation run time was mostly within 30 min. For 3D simulations that ran until failure, each simulation took about half a day. Figure 1 is an example of the RVE geometry of the model with randomly distributed core–shell nanoparticles.

In the RVE model, there were 30 randomly distributed core–shell nanoparticles to maintain a weight fraction of 2% with an RVE length of 186 nm. The periodicity of the model was ensured by generating the same particle constellation on the opposite surfaces of the RVE. Due to the complexity of the geometry, the matrix and core–shell nanoparticles were meshed with first-order tetrahedral elements (C3D4). Static analyses were carried out with Abaqus/Standard. Different material phases were modeled separately, and the nodes were tied together. There was a total of 527,919 elements in the entire model. For each nanoparticle, the spherical core had 177 elements, and the shell had 455 elements. The brittle failure criterion was applied to the matrix. When the criterion was satisfied, the corresponding elements were deleted as it could no longer take any load. The epoxy matrix was assumed to behave as an isotropic linear elastic solid. The Rankine criterion was applied to detect damage initiation. Damage initiates when the maximal principal stress in the RVE exceeds the tensile strength specified in the modeling.
(1)$$\sigma \geq \sigma_{t}|\;\sigma >0,$$
where σ is the maximal principal stress experienced in the RVE, and $\sigma_{t}$ is the tensile strength of the material.

A displacement boundary condition was applied to the nodes on the surfaces of the RVE. Uniaxial tension was applied in the *x* direction. The lengths of the RVE model are denoted as $L_{x}$, $L_{y}$, and $L_{z}$, and the displacement boundary condition can be expressed as:(2)$$\begin{array}{c} u\left(0,y,z\right)=0,\\ v\left(x,0,z\right)=0,\\ w\left(x,y,0\right)=0,\\ u\left(L_{x},y,z\right)=\delta_{x}, \end{array}$$
where $\delta_{x}$ is the prescribed displacement for loading along the *x* direction. For this specific problem, the macroscopic nominal stress and strain can be obtained as follows:(3)$$\begin{array}{c} \sigma_{x}=\frac{F_{x}}{L_{y}\times L_{z}},\\ \varepsilon_{x}=\frac{\delta_{x}}{L_{x}}, \end{array}$$
where $F_{x}$ is the reaction force on the loading surface.

Upon applying uniaxial tension to the RVE, the simulated composite strength was at about the same level as that of the matrix strength, and no strengthening effect was observed. The damage pattern is shown in Figure 2. The colored contour in Figure 2 represents the scalar degradation status (SDEG), which is an indicator of the damage variable based on the maximal stress that a damaged element can sustain. When the SDEG value of any element reaches unity for cohesive elements, and 0.99 for other type of elements, this element is fully damaged and is deleted, as it cannot sustain any loading. A cross-sectional damage process is shown in Figure 2 with the loading applied in the horizontal direction with respect to the plane. Figure 2a is the undamaged state. Figure 2b shows that the damage (cracks) started to appear at a few separate locations at the matrix regions around the nanoparticles. As the damaging process progressed, nearby cracks eventually joined up to form a through crack and caused catastrophic failure to the nanocomposite, as shown in Figure 2c.

To simplify the system, a single core–shell nanoparticle placed at the center of the RVE was considered instead of multiple randomly distributed nanoparticles in the previous model. The nanoparticle size was kept the same, while the RVE length was modified such that the weight percentage of the nanoparticle was maintained at 2%. The RVE length was 60 nm based on the properties and nanoparticle content. The meshing is shown in Figure 3, and the damage progression is shown in Figure 4. Figure 5a is the stress–strain responses when uniaxial tension was applied to the model. Pure matrix refers to the same RVE geometry with the materials of all sections set as the matrix. The responses of RVEs with both a single nanoparticle and multiple randomly distributed nanoparticles embedded are presented in Figure 5a. The similarity in the plots, especially for loading up to the point of maximal load, indicates that the simplification of multiple nanoparticles to a single nanoparticle gave a reasonable prediction of the mechanical response of such nanocomposites.

Having established that a simplified single-particle model gives a reasonable prediction of the strength and stiffness of the nanocomposites, the model was used to investigate the effects of different particle concentrations. Figure 5b shows the stress–strain responses of RVEs with a single nanoparticle of different weight percentages. The difference mainly lay in the poststrength behavior. The resulting Young’s moduli are listed in Table 2. The stiffness increased from 2.36 to 2.52 and 2.58 GPa after the addition of 1% and 2% (weight percentage) nanoparticles, respectively. The improvement matches with the results in [15]. Different from some of the experimental results, the tensile strength of the nanocomposite barely reached the strength value of the pure matrix.

Figure 6 indicates the corresponding cross-sections at various points on the stress–strain curves. Tensile strength was reached immediately after damage had started to appear, as shown by Label 1 in Figure 6. As the damage process continued, the stress began to drop, as the material could not take as much load, as indicated by Labels 2 and 3.

To further improve the modeling, a very thin layer (1/200 of the nanoparticle diameter) of the cohesive elements was then inserted into both the core–shell and shell–matrix interfaces to mimic the debonding between different materials since, in the previous modeling, perfect bonding was assumed. The density of the cohesive layers was assumed to be the same as that of the matrix. The damage of cohesive layers, i.e., debonding, is characterized by a traction-separation law, also known as the cohesive zone model, which is often utilized in modeling delamination. In the current model, the maximal stress criterion is selected for damage initiation. It can be expressed as:(4)$$max\;\langle \frac{t_{n}}{\sigma_{n}^{c}},\frac{t_{s}}{\sigma_{s}^{c}},\frac{t_{t}}{\sigma_{t}^{c}}\rangle =1,$$
where $t_{n}$ is the nominal traction stress component along the normal direction and $t_{s}$, $t_{t}$ are the tractions in two shear directions; $\sigma_{n}^{c},\;\sigma_{s}^{c},\;\sigma_{t}^{c}$ are the cohesive strengths of the interface. Macaulay brackets 〈 〉 are used to represent the condition that failure only occurs under positive stress. In this study, the cohesive strengths are assumed to be equal in all directions. The experimentally measured tensile strength of epoxy is about 50 MPa, as stated in [14,15]. This value serves as a reference for interfacial strengths. Different combinations of core–shell and shell–matrix interfacial strength values are investigated, and the resulting tensile strengths are listed in Table 3.

Table 3 shows the combinations of core–shell and shell–matrix interfacial strength values that were simulated. All combinations gave much lower strength than the pure matrix strength of 50 MPa due to the stress concentrations when inhomogeneity is introduced. For a fixed core–shell interfacial strength value, a lower (than the matrix strength) value of shell–matrix interfacial strength reduced the overall strength to a greater extent. Once the shell–matrix strength surpasses the matrix strength, the nanocomposite strength would not be further affected by the shell–matrix interface. On the other hand, the overall tensile strength of the composite was not affected by the core–shell interfacial strength as long as the shell–matrix strength was fixed.

Figure 7 is a cross-section of the stress distribution where the shell–matrix interfacial strength was 40 MPa, lower than the matrix strength, and the core–shell strength was 60 MPa. Uniaxial tension was applied in the horizontal direction (referring to Figure 7b). Since the shell–matrix interface was weaker, it started to fail before any other part of the nanocomposite, resulting in debonding between the inclusion and the matrix as shown in Figure 7a. Due to the inhomogeneity of the materials, stress concentrations appeared between the shell and matrix at the top and bottom of the inclusion. As the stress further increased and reached the strength of the matrix, damage started to appear and propagate in the matrix, as shown in Figure 7b. Due to this stress concentration, the nanocomposite was then only able to bear a lower load compared with the pure matrix.

Figure 8 illustrates the case where the core–shell interface was weaker than both the matrix and shell–matrix interface. Though debonding between core and shell happened first, catastrophic failure still occurred in the matrix part due to its lower strength, starting near the shell due to the stress concentrations.

## 3. Nanocomposites with Initial Crack

The theoretical strength of most thermoset polymers can be calculated on the basis of the bonding between polymer chains. The calculated strength value is about 10% of the Young’s modulus, which falls in the range of 200 to 400 MPa for most thermoset polymers, including epoxy, silicone, and polyurethane [39,40]. However, the experimentally measured strength value was much lower than this range. Considering that defects may naturally occur or be introduced into a material at various stages, such as synthesizing and manufacturing, the idea is proposed of stress concentrations at the defects being reduced with the addition of nanoparticles, with a consequent increase in material strength. Cracks were selected as the initial defects, as they result in an extremely high stress concentration/intensity. To have a clearer illustration and better control of parameters, two-dimensional (2D) models were initially investigated. Since the locations of defects are random, the defect was placed at various positions within the 2D model in order to have a holistic view. Some nondestructive testing (NDT) methods for defect detection have been developed in specific areas such as aeronautics [41], and the technique of defect detection in other fields is not as widely applied. Therefore, in this work, the initial crack was placed at designated positions that covered a significant portion of the model, such that a representative contour figure could be plotted. Parameters such as initial crack length and matrix-to-particle stiffness ratio were also studied.

As the maximal stress in the matrix is the main concern here, the failure of the core and shell was not considered. Therefore, the core–shell structured nanoparticle is represented as a homogenized single-phase particle of the same size with twice the stiffness of the matrix, and perfect bonding was assumed between particle and matrix. A validation test was performed on a three-dimensional (3D) RVE to check that this simplification did not affect the stress distribution in the matrix, especially the maximal stress. Figure 9a shows cross-sectional stress distribution in the matrix of the RVE with a core–shell nanoparticle; in Figure 9b, this core–shell inclusion was replaced by a homogenized single-phase nanoparticle with twice the stiffness of the matrix. The nanoparticles were removed from both figures for a clearer view of stress distributions in the matrix. The stress magnitudes and distributions did not differ significantly between the two RVEs. In addition, the stiffness improvement resulted from the homogenization matching with the experimental work in [15].

Studies in 2D were first carried out to provide some guidance for 3D computational analysis, as 2D models are computationally less time-consuming. A pre-existing crack was inserted into the model. The 2D model may be viewed as a cross-sectional cut from the midplane of a 3D RVE. Figure 10 is an example of a 2D plate with a homogenized nanoparticle at the center, and a 5 mm crack located at its top-right quarter. The diameter of the nanoparticle was 18.8 nm, as stated in [14].

The stiffness of the homogenized nanoparticle was set to be twice that of the matrix. The nanoparticle was sited at the center of the RVE together with an initial crack at various locations. The length of each side is set as 60 nm, same as the 3D RVE length. Uniaxial tension was applied in the horizontal *x* direction. Two different crack lengths, 2 and 5 nm, were investigated. Due to symmetry, the locations of the crack needed only to be varied within the top-right quarter of the plate. Linear elastic material properties were assumed to be only the highest stress values at a fixed strain and were compared.

## 4. Effects of the Positions of the Initial Crack

Two simulations were performed for each fixed crack position, one with an entire matrix property, and the other with an inserted nanoparticle. The stress experienced at the bottom end of the crack, which is always where the maximal stress occurs, was considered. Normalized values, calculated as the ratio of the maximal stress of the nanocomposite RVE to that of the pure matrix with a similar crack, were used for analysis.
$$\mathrm{Relative}\;\mathrm{stress}=\frac{\sigma_{max}^{RVE}}{\sigma_{max}^{matrix}}$$

Figure 11a is the top-right quarter of a 2D plate with a homogenized nanoparticle at the center. The empty bottom-left squares represent a quarter of the particle. The numbers denote the relative stress of the bottom crack tip in RVE normalized by that of pure matrix model (50 MPa). The location of each relative stress value in Figure 11a is also the location of the bottom tip of the crack. Figure 11b is the contour plot of Figure 11a.

The band of highlighted numbers in Figure 11a was inclined at 45° from the horizontal. Above this 45° line, the normalized value was mostly below 1, indicating that the presence of the nanoparticle caused the maximal stress to decrease as compared to that of the pure matrix model. A decrease in maximal stress in turn allows for the overall structure to bear greater stress, and thus increases the tensile strength. As the crack moved away from the nanoparticle, the effect of the nanoparticle was reduced. Below the 45° line, on the other hand, there was an increase in relative stress values. As a result, whether an inserted nanoparticle strengthens or weakens the overall material is dependent on the location of the defect relative to the nanoparticle and direction of loading. This could be one of the possible reasons why some studies showed improvement in tensile strength while the others did not, as the location of a natural or initial defect is difficult if not impossible to control.

Figure 12 is also the contour plot of a plate with an initial crack, where the crack length was 2 nm instead of 5 nm, as in Figure 11. The trend was about the same, except for the exceptionally high value that was directly to the right of the nanoparticle.

For an inclusion that is stiffer than the matrix, the normalized maximal stress was the smallest and always less than unity when the crack was situated above the inclusion, i.e., when the inclusion and crack were aligned to the direction of unloading. Conversely, the normalized maximal stress was the greatest and always greater than unity when the relative positions of the inclusion and crack were perpendicular to the direction of unloading. For situations where the relative positions of inclusion and crack were 45° to the loading direction, the normalized maximal stress is close to 1. Additionally, when the inclusion and crack were aligned to the direction of unloading, such as the case shown in Figure 10, the inclusion behaved like an obstacle that directly blocked the crack from growing. In order to propagate, the crack needs to change its propagating path by bending around the inclusion. More energy is required to achieve such bending. As a result, the nanocomposite is then able to bear with a higher stress to achieve the same amount of crack growth.

Figure 13, Figure 14 and Figure 15 show the normalized maximal stress contours for nanoparticles of 3 different stiffness ratios relative to the stiffness of the matrix 0.5, 2, and 5. A crack of 5 nm in length was taken as the initial defect. When the nanoparticle was stiffer than the matrix, there was a reduction in normalized maximal stress when the crack was above the 45° band, i.e., perpendicular to the direction of loading. The greatest reduction occurred when the crack was directly on the top of the nanoparticle. Below this 45° line, there was mostly negative or nonsignificant impact. In addition, if the relative stiffness was further increased, the general observation was similar, but the extent of improvement or deterioration increased. However, when the nanoparticle was less stiff than the matrix, the observed trend was the opposite. Stress reduction was achieved for cracks below the 45° line, i.e., towards the loading direction, with the greatest reduction observed when the crack was directly to the right of the nanoparticle.

## 5. Three-Dimensional Validation

A few three-dimensional RVEs were created to verify that the trends observed in 2D could be observed in 3D as well. For the 3D models, the initial defect is an elliptical crack. The sharpness of the crack tip can be controlled by adjusting the aspect ratio of the ellipse. The length of the major axis was the same as the crack length (5 nm) in the 2D case, and the minor axis was set to be one-quarter of the major axis length.

As observed from 2D simulations, the RVE could be categorized into two regions—one for crack locations that increased the strength of the nanocomposite, and the other where the strength was decreased depending on the loading direction. RVEs with the elliptical crack located at each region were created, and maximal stresses around the crack were computed. The investigated cases are summarized in Table 4. All the RVEs were loaded along the x axis following the coordinate system shown in Table 4. In the first RVE, the crack was placed to the side of the nanoparticle when observing from the x direction, along an axis perpendicular to loading corresponding to the favorable region in the previous 2D model. The relative maximal stress, calculated by the maximal stress experienced by the RVE divided by that experienced by the pure matrix model, also indicated that the presence of the nanoparticle was beneficial to the strength of the nanocomposite. The elliptical crack in the other RVE was placed along the loading direction, corresponding to the unfavorable region in 2D modeling. A detrimental effect to the strengths of the RVEs was observed. Results of the 3D RVEs are in line with the 2D models, indicating that the preliminary studies using simplified 2D models are valid.

## 6. Conclusions

On the basis of the experimental work of adding a layer of rubber coating to the nanoparticles when fabricating nanocomposites, a three-dimensional computational model with randomly distributed nanocomposites was created. The strength enhancement that appeared in some of the experiments could not be repeated in finite element simulations. The new idea of introducing initial precracks into the system was proposed, considering that materials are not perfect in nature. The stress at the crack tip of matrices with an embedded nanoparticle to the stress at the crack tip of matrices without nanoparticles was computed for different crack locations. There existed a dividing line along the 45° direction. If the relative positions of the inclusion and the crack were away from the loading direction, i.e., the initial defect was located above the dividing line in this case, the added nanoparticle positively impacts the maximal stress reduction and thus enhances the overall tensile strength. On the other hand, if the initial defect was located below this line (the defect and the inclusion were aligned to the loading direction), the effect was reversed. Parametric studies showed that the initial crack length did not seem to be a very critical parameter as the relative stresses for the studied quadrant, showing a similar distribution regardless of crack length values. The relative stiffness of nanoparticle to matrix is a more dominant factor. For RVEs where the nanoparticle was stiffer, increasing the relative stiffness did not alter the distribution, but the degree of strength enhancement or deterioration was also magnified. On the other hand, if the nanoparticle was less stiff than the matrix, the trend was the opposite. The region towards the loading direction (below the 45° line) then became favorable for initial crack locations. Validation tests were performed on 3D RVEs with an elliptical crack as the initial defect. The results from 3D models matched with the 2D findings, i.e., an elliptical crack placed along the loading or nonloading directions led to an increase or reduction in maximal stress, respectively.

## Figures and Tables

**Figure 1 materials-15-04906-f001:**
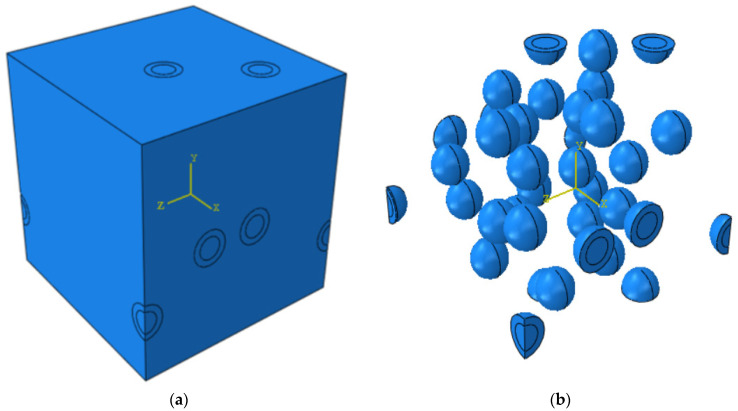
An example of (**a**) a nanocomposite RVE and (**b**) core–shell nanoparticles.

**Figure 2 materials-15-04906-f002:**
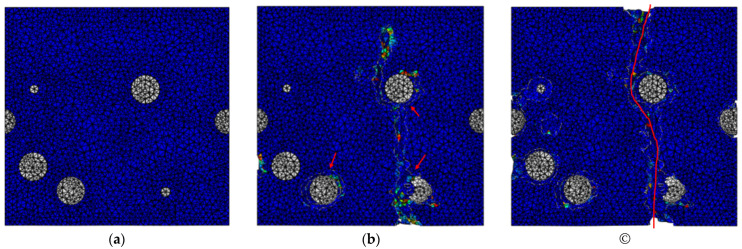
Damage progression in nanocomposite: (**a**) before damage; (**b**) onset of damage; (**c**) catastrophic failure. (Red arrows and line indicate damage sites.)

**Figure 3 materials-15-04906-f003:**
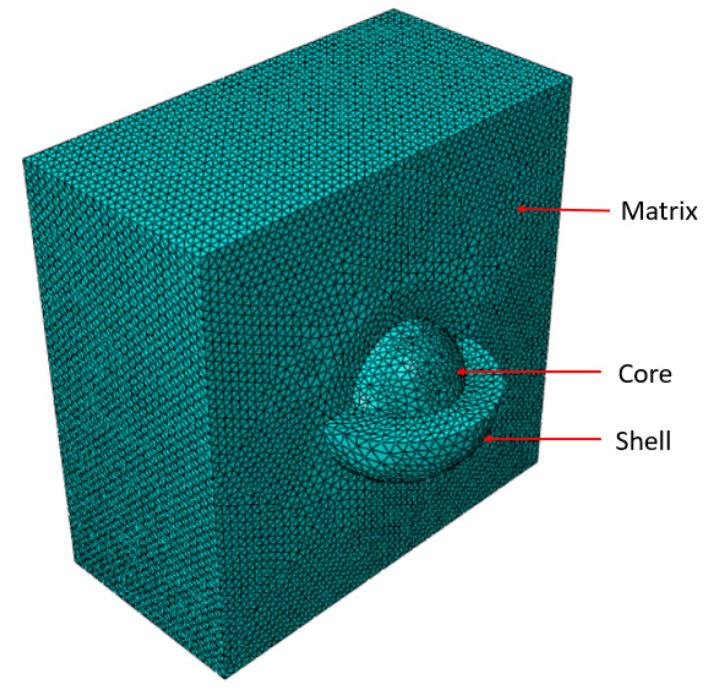
Meshing of the various parts of an RVE.

**Figure 4 materials-15-04906-f004:**
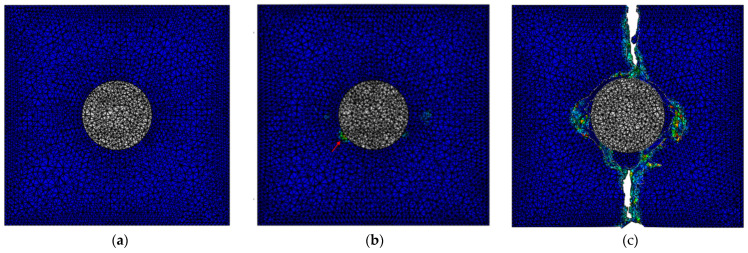
Damage progression in nanocomposite: (**a**) before damage; (**b**) onset of damage; (**c**) catastrophic failure. (Red arrow indicates the damage site).

**Figure 5 materials-15-04906-f005:**
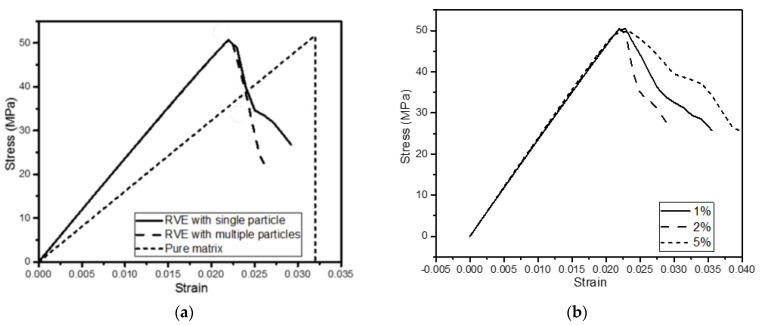
Stress−strain curves of (**a**) matrix and RVEs with single and multiple nanoparticles; (**b**) RVEs with a single nanoparticle of different weight percentages.

**Figure 6 materials-15-04906-f006:**
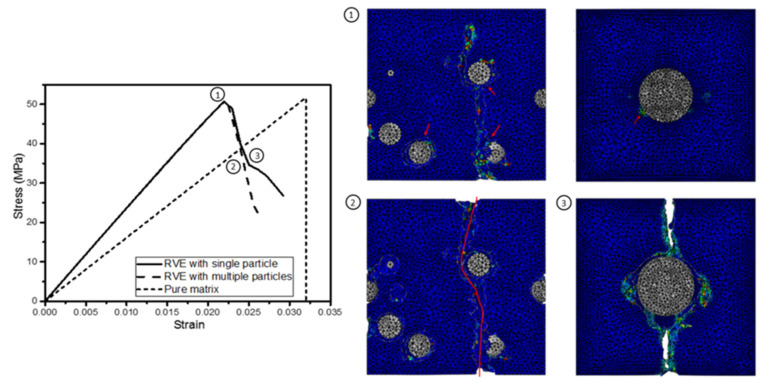
Cross-sections at different stages of stress–strain curves. (Red arrows and line indicate damage sites).

**Figure 7 materials-15-04906-f007:**
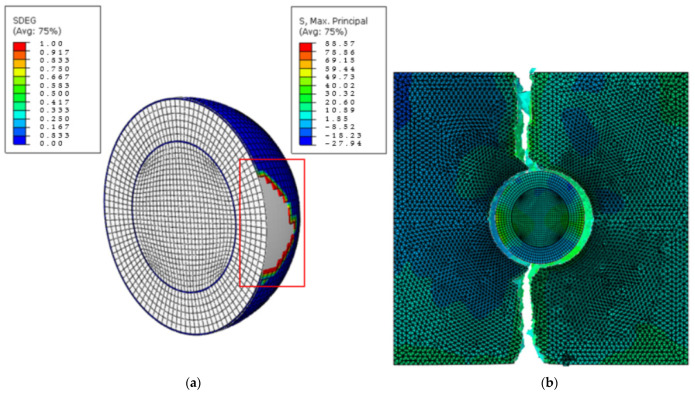
Cross−sectional images of (**a**) nanoparticle and (**b**) nanocomposite damage for core−shell strength = 60 MPa and shell−matrix strength = 40 MPa. (The damage site is indicated by the red frame and the unit for stress is MPa).

**Figure 8 materials-15-04906-f008:**
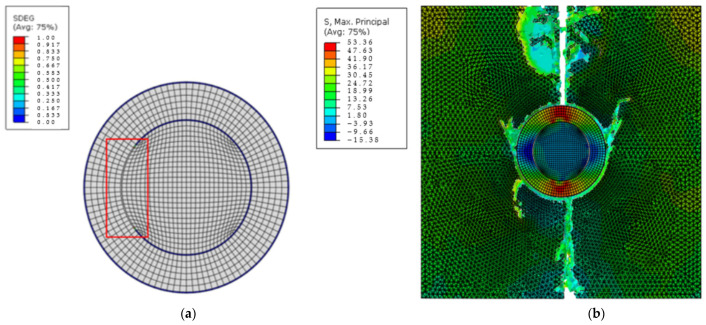
Cross-sectional images of (**a**) nanoparticle and (**b**) nanocomposite damage for core–shell strength = 40 MPa and shell–matrix strength = 60 MPa. (The damage site is indicated by the red frame and the unit for stress is MPa).

**Figure 9 materials-15-04906-f009:**
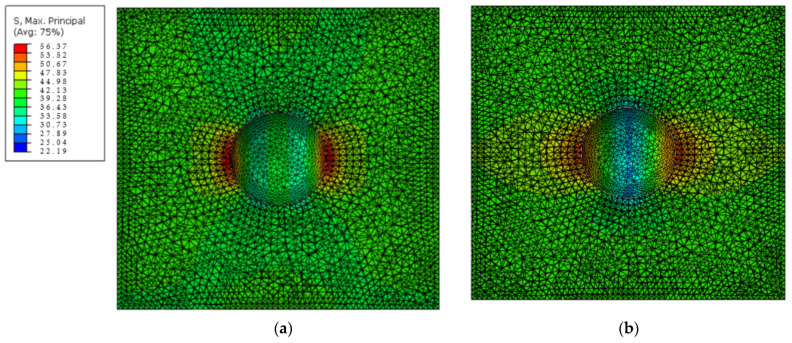
Cross-sectional stress distribution in the matrix of the RVE with (**a**) one core–shell nanoparticle; and (**b**) one homogenized single-phase nanoparticle. (The unit for stress is MPa).

**Figure 10 materials-15-04906-f010:**
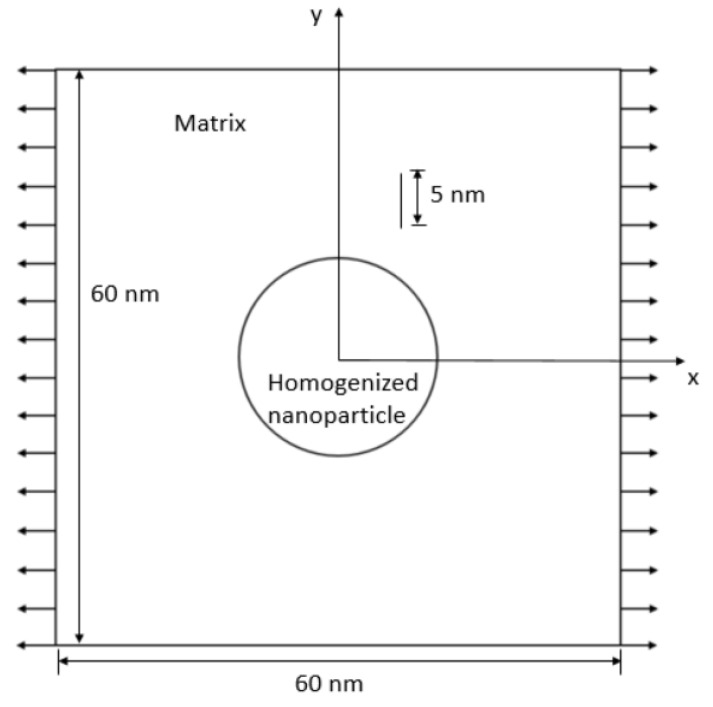
A 2D RVE with nanoparticle and 5 nm crack.

**Figure 11 materials-15-04906-f011:**
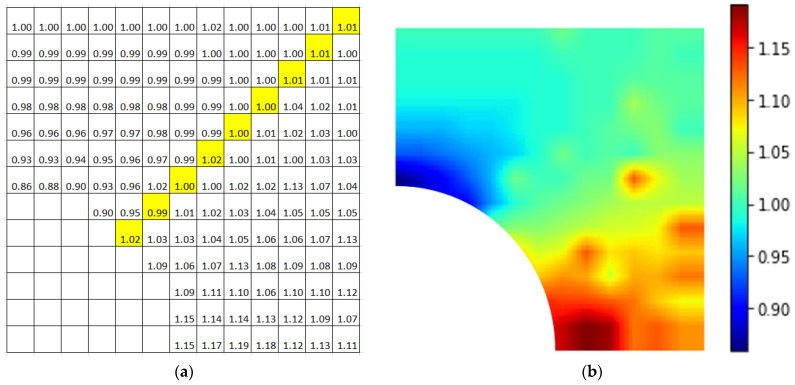
Plot of RVE with an initial crack (crack length = 5 nm) (**a**) with normalized maximal stress values; (**b**) contour plot. (The highlight is inclined at 45° from the horizontal).

**Figure 12 materials-15-04906-f012:**
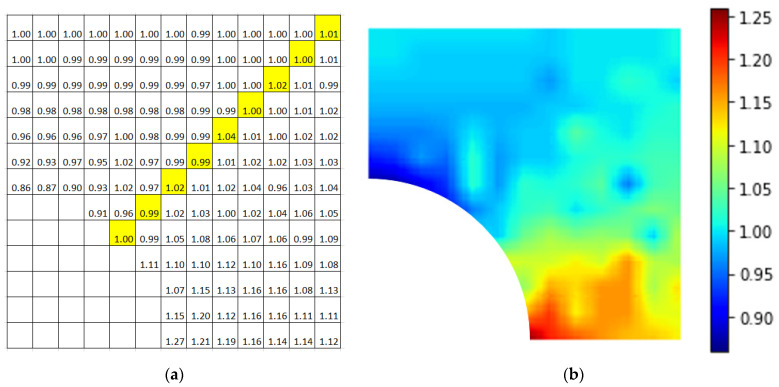
Plot of RVE with an initial crack (crack length = 2 nm) (**a**) with normalized maximal stress values; (**b**) contour plot. (The highlight is inclined at 45° from the horizontal).

**Figure 13 materials-15-04906-f013:**
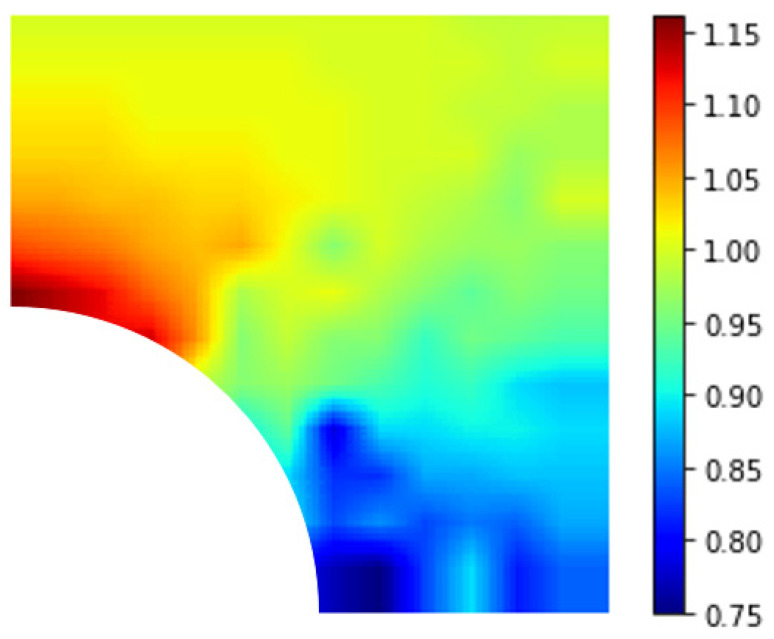
Contour plot of RVE with an initial crack where $E_{p}/E_{m}=0.5$.

**Figure 14 materials-15-04906-f014:**
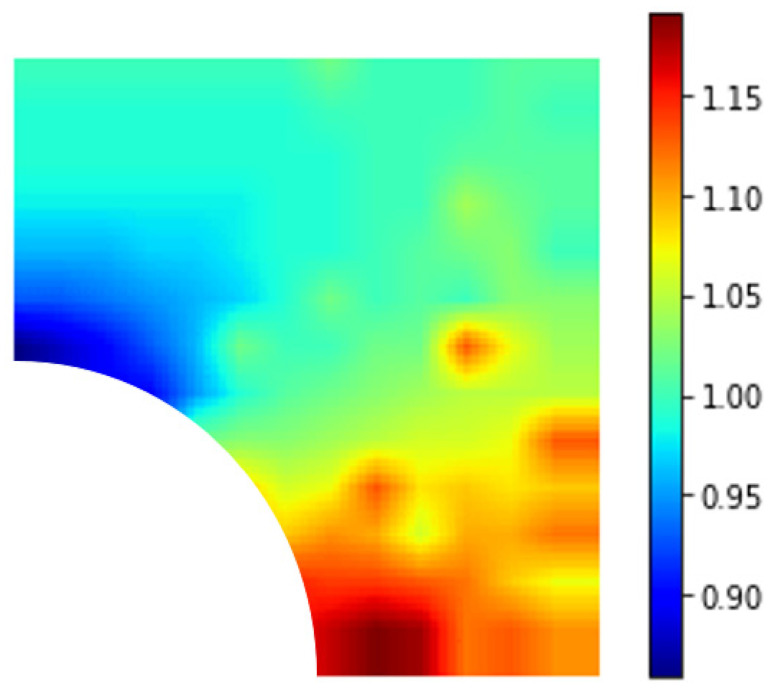
Contour plot of RVE with an initial crack where $E_{p}/E_{m}=2$.

**Figure 15 materials-15-04906-f015:**
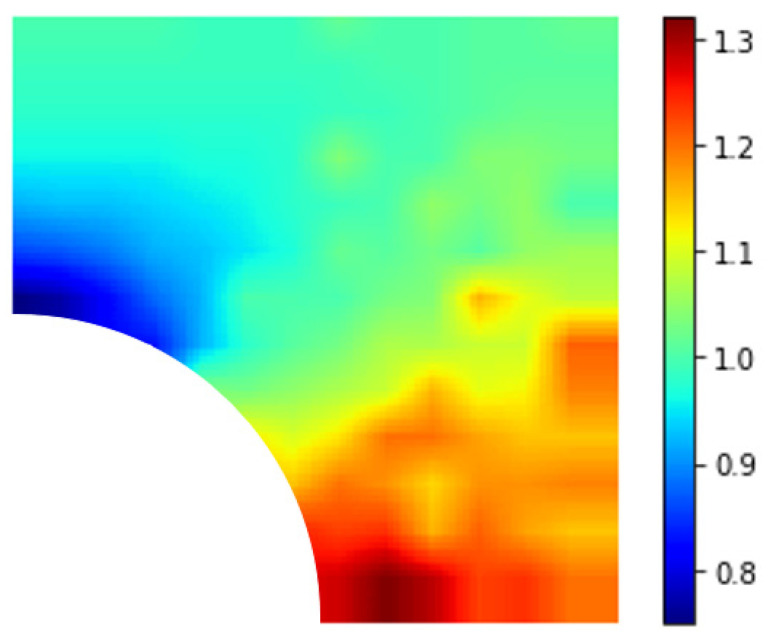
Contour plot of RVE with an initial crack where $E_{p}/E_{m}=5$.

**Table 1 materials-15-04906-t001:** Properties of various parts of the RVE.

	Young’s Modulus (GPa)	Poisson Ratio	Failure
Matrix	2.4 [14]	0.3	Fracture energy = 0.001 N/m
Core (nanosilica)	75 [38]	0.17	-
Shell	2.6 [35]	0.3	-

**Table 2 materials-15-04906-t002:** Stiffness of RVEs with different nanoparticle contents.

	Pure Matrix	1% Nanoparticles	2% Nanoparticles	5% Nanoparticles
E (GPa)	2.36	2.52	2.58	2.72

**Table 3 materials-15-04906-t003:** Values of core–shell and shell–matrix interfacial strength tested.

	Core–Shell Strength (MPa)	Shell–Matrix Strength (MPa)	Core–Shell Strength/Shell–Matrix Strength	RVE Strength
Combination 1	40	40	1	21 MPa
Combination 2	40	60	0.667	27 MPa
Combination 3	60	40	1.25	21 MPa
Combination 4	60	60	1	28 MPa
Combination 5	60	80	0.75	28 MPa
Combination 6	60	2000	0.03	28 MPa
Combination 7	70	60	1.17	28 MPa
Combination 8	80	60	1.33	28 MPa
Combination 9	90	60	1.5	28 MPa
Combination 10	2000	60	33.3	28 MPa

**Table 4 materials-15-04906-t004:** 3D RVEs with ellipsoid hole as the initial defect.

RVE	Relative Maximal Stress
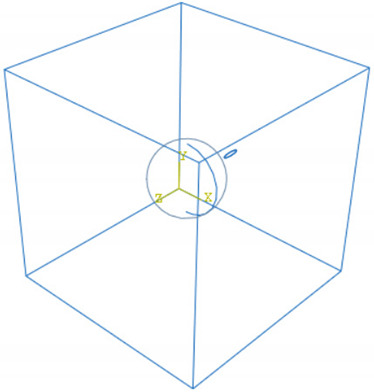	0.92
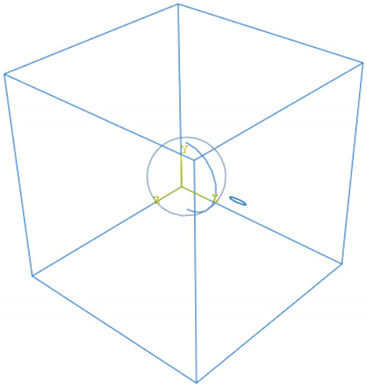	1.18

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
