# Peer review of "Effects of Relative Positions of Defect to Inclusion on Nanocomposite Strength"

_materials, 2022, doi:10.3390/ma15144906_

Round 1
Reviewer 1 Report
The submitted article " Effect of relative positions of detect to inclusion on Nano- composite strength, woks on systematically generated defects using the Abaqus software.
The article is well written and has interesting findings. However, I have two-three minor issues described here.
Page 1, lines 36-37, reference to this statement is missing.
Line 38-39, the justification of this statement is missing.
Discussed literature review and previous work described are outdated. Recent research activities should be included for the article's soundness. Overall, references to recent information should be included in this article.
Reviewer 2 Report
The paper is well written and well organized. However, there are some points which need clarification/correction. My comments on the paper are as follows:
1) The results presented in Fig. 5 suggest that nanocomposite modulus is independent of nano filler concentration. This is not observed in the results presented by other researchers. The authors should provide reasons for this.
2) The data presented in the last column of Table 2 are unclear “RVE Strength Compared with Matrix” is unclear. Please use a more clear title for the column.
3) There are many English/wording mistakes throughout the paper, some are:
a. Page 1, line 21: “outcomes” should read “results”.
b. Page 2, Line 48: “displays a higher value” should read “display higher values”.
c. Page 2, Line 50: “resin only” should read “pure resin”.
d. Page 2, Line 61: “elongation at break” should read “elongation to failure”.
e. Page 2, Line 72: “strength value of a pure” should read “strength of pure”.
f. In Eq. 1; Placing of σ1˃ 1 is confusing. Please correct.
And many more such mistakes
Reviewer 3 Report
In this study, the authors present a parametric finite element (FE) analysis to quantify the effects of relative positions of defect to inclusion on the strength of core-shell rubber nano-fillers reinforced composites. The numerical study is performed using Abaqus commercial FE software. The numerical computations are performed on 2D and 3D representative volume elements (RVEs). By including a pre-existing crack in the matrix of the RVE, the authors showed that the stress concentration at the crack tip is reduced for cases where the nanoparticle and pre-crack are aligned away from the loading direction. The stress concentrations around inherent defects in materials can therefore be reduced by adding nanoparticles to improve material strength. However, it is shown that the inverse effect is obtained if the crack and nanoparticle are aligned towards the loading direction. Besides, the authors performed parametric in terms of relative stiffness of nanoparticle to matrix and crack length. The results reported in this paper are interesting. The paper is interesting from numerical point of view and the topic of interest to the readers of Materials. The paper is well written and well structured. Therefore, it can be accepted in the present form.Author Response
Thank you for your kind comments.
Reviewer 4 Report
1- It is interesting to show the effect of relative defect to inclusion position from energy point of view, before and after crack propagation. Also it is stated in abstract, but it is not discussed.
2- It is suggested to mention the CPU-run-time and computer hardware specifications.
3- ABAQUS element type, analysis step type, interaction properties, dynamic loading rates and the effect of temperature should be explained enough.
4- No verification of the results is observed within the manuscript.
5- Please cite related works such as the following article:
Gong, Yanfeng, et al. "A deep transfer learning model for inclusion defect detection of aeronautics composite materials." Composite structures 252 (2020): 112681.
Reviewer 5 Report
The research area is: Effects of Relative Positions of Defect to Inclusion on Nano-composite Strength How was randomness created in the model? Has any simulation technique been used? How are the input parameters selected for the calculation? It would be more appropriate to perform your own experiments. Is the given fracture energy a representative value? Figure 6 - clearly state the dimensions of the modelled area? How many finite elements did the model have? What parameters are used for the calculation? must provide all relevant information. The calculations and results are interesting, but they need to be commented on and described in order to increase the credibility of the information and the information value. The chosen concept of numerical modelling is interesting, but the detail and scope of the performed calculations have a limited informative value. Overall, it is necessary to better specify the problem to be solved in a clear connection to experimental research. To solve the micro-level, it is necessary to specify in more detail the inclusion of material, structural and geometric uncertainty and nonlinearity. References are not in MDPI template format. The manuscript has no part of the discussion. There is no comment on the results of the manuscript and information of the current state of affairs research. One of the weaknesses of the manuscript is that it has little connection to experimental and materials research. The manuscript should be completely reworked.Author Response
Please see the attachment.

Round 2
Reviewer 4 Report
Dear respected author,
The replies to the comments are satisfactory. It is recommended for publication in its present form.
Regards
Reviewer 5 Report
The manuscript has been improved.
Thanks for the edits.